# Moderate Cardiovascular Exercise Speeds Up Neural Markers of Stimulus Evaluation During Attentional Control Processes

**DOI:** 10.3390/jcm8091348

**Published:** 2019-08-30

**Authors:** Axel H. Winneke, Lena Hübner, Ben Godde, Claudia Voelcker-Rehage

**Affiliations:** 1Project Group Hearing, Speech and Audio Technology, Fraunhofer Institute for Digital Media Technology, 26129 Oldenburg, Germany; 2Institute of Human Movement Science and Health, Chemnitz University of Technology, 09126 Chemnitz, Germany; 3Department of Psychology & Methods, Jacobs University, 28759 Bremen, Germany

**Keywords:** EEG, ERP, Flanker task, attentional control, acute cardiovascular exercise

## Abstract

Moderate intensity cardiovascular exercise appears to provide a low-cost “intervention” on neurocognitive processes such as attentional control, yet the effects vary depending, for example, on cognitive task, time of testing, or exercise intensity. However, while a number of studies show that brief bouts of acute exercise can modulate behavioral indices of cognitive control, relatively few studies have attempted to identify the brain activity associated with these changes immediately following exercise. Here, we tested 11 young adults in a crossover design with a Flanker task at rest and immediately (within 2–3 min) following 20 min of acute exercise at 60% of the individual VO_2_max. In order to prevent delayed exercise effects that might confound or dilute immediate effects, a short version of the Flanker task (8 min) was chosen and an EEG was recorded simultaneously. The N2 and P3 ERP components were analyzed in addition to accuracy and response time. The N2 reflects conflict resolution, and the P3 has been linked to stimulus evaluation processes. No effect of exercise was found for behavioral data but P3 peak latencies were shorter following exercise as compared to rest. The N2 amplitude data suggest that exercise seems to prevent a decline in resources of attentional control over time. These data indicate that acute exercise, at a moderate intensity level, speeds up neural processing of attentional control by modulating stimulus evaluation processes immediately following exercise and that exercise helps maintain a steady level of neurocognitive resources.

## 1. Introduction

Although there is evidence for positive effects of acute exercise on cognition (for reviews cf. [1,2,3,4,5]), there is no general consensus likely due to heterogeneity in study designs (cf. [6]). A meta-analysis by Lambourne and Tomporowski [7] indicates that cognitive performance following exercise improves on average. A possible explanation for the benefit on performance is that acute exercise activates attentional control resources [8,9,10,11]. The finding that brief, moderate-intensity exercise can improve performance in subsequent cognitive tests has important implications. For example, a study involving adolescents in a school setting revealed increases in attentional capacities following a 10-minute bout of physical, coordinative exercise, thereby supporting the argument that physical activity could help to improve scholastic performance and cognitive functioning [12,13,14]. This is in line with results from a meta-analysis [15], which suggests a beneficial effect of acute, moderate-level physical interventions on cognition including attentional control in younger adults. Even though several event-related potential (ERP) studies have looked at effects of an acute bout of exercise on cognition, the time window in which the effects have been evaluated vary considerably. Most studies conduct experimental cognitive tasks with some delay (e.g., 48 min [16], 40 min [11], 20 min [17,18], 15 min [19], 16 min [14]). Only few studies start ERP recordings immediately after exercising (within 5 min) but use different cognitive tasks [8,20,21,22] and report inconsistent results. See Pontifex and colleagues ([23] supplementary table) for a recent review that shows the paucity of ERP studies on the immediate effects (0–5 min after exercise cessation) of acute exercise on the inhibitory system. Thus, the aim of the current study was to further our understanding of these immediate effects of moderate-intensity exercise on attentional control as measured by a Flanker paradigm and associated neural mechanisms. ERPs were recorded to better understand the underlying (neural) processes of the relationship between acute exercise and cognitive functioning.

The Flanker task [24] taps into attentional control as it requires the ability to focus attention on a predefined target while inhibiting the distracting influence of irrelevant stimuli [25,26]. The neurocognitive processes involved in the Flanker task are often derived by means of ERPs such as the N2 and the P3. The N2, which is a negative deflection occurring about 170–250 ms after stimulus onset, is said to predict successful resolution of conflicts caused by the interference of distracting stimuli; the larger the amount of attentional resources devoted to conflict resolution between target and distracting stimulus, the larger the amplitude [27]. The N2 is maximal at frontal locations and likely to originate from the anterior cingulate cortex. The P3 is a positive deflection starting about 300 ms after stimulus onset with neural generators located in frontal and temporo-parietal regions [28]. While the P3 amplitude is an index of allocation of available resources, P3 latency is considered to reflect the timing of mental processes [29]. P3 amplitude has been shown to decline with task difficulty, including incongruent flanker trials (e.g., [29,30,31,32]). According to Daffner and colleagues [33], a reduction in P3 amplitude with increasing task difficulty is related to fewer available resources. During conflict trials of the Flanker task, P3 latencies are usually delayed, which is indicative of a prolongation of stimulus evaluation processes [34,35,36].

Results regarding the effect of exercise on the N2 are few and mixed. Using a Flanker task, Themanson and Hillman [11] did not find effects of acute exercise on the N2 after exercising whereas Pontifex and Hillman [37], also using a Flanker task, reported a reduction in N2 amplitude while exercising (cf. [38] for children). Using a Stroop task following moderate exercise, Chang and colleagues [19] also reported no effects on the N2. Given the diversity of results, the effect of exercise on the N2 as a marker of attentional control requires further investigation.

Effects of acute exercise on P3 amplitudes and latencies have been described more frequently. Acute bouts of exercise led to increased P3 amplitudes and shorter peak latencies alongside behavioral benefits in cognitive attention tasks such as oddball or Flanker tasks (e.g., [8,16,38,39,40]). Using a Flanker task, Hillman and colleagues [16] recorded ERPs after participants had returned to their baseline heart rate (HR) (on average 48 min) after a high-intensity acute exercise intervention (30 min at sub-maximal intensity, i.e., 83% of HRmax). The intervention led to reduced P3 latencies and increased P3 amplitudes subsequent to exercising indicating an increase in attentional resources capacity (e.g., [9]). Also, other studies looking into the effect of acute bouts of moderate-intensity exercise on attentional control reported similar results [17,18,19,41,42], however using different cognitive tasks (Flanker, Go/No-Go, Attention-Network-Task, Stroop) and investigating different time windows post exercise; for example, 20 min after [17,18], or 10 min after but a task duration of 25 min [41]. Further support for an enhancing effect of an acute bout of exercise on attentional resources is added by studies looking at different tasks such as a Go/No-Go task [22,43] or different brain measures such as event-related desynchronization of the EEG alpha frequency band [44].

Although a substantial amount of research has investigated acute effects of exercise, the majority commenced experimental testing after delayed time intervals, and the question as to the immediate effects of exercise on attentional processes and its neural underpinnings is still not thoroughly researched. Only a limited number of studies commenced experimental EEG recording immediately after (i.e., within 5 min) exercise but employing different cognitive tasks [8,20,21,22,43]. Accordingly, the results vary for these studies with some showing behavioral effects of exercise [8,21] but others not [20,22,43]. Similarly, Kamijo and colleagues [8,21,43] report larger P3 amplitudes and shorter P3 latencies following exercise [8,21]; Pontifex et al. [20] observe no effect on amplitude but a slowing of P3 latencies when comparing before to after exercising as well as when comparing before to after sitting. When comparing post-sitting vs. post-exercising, a decrease in P3 amplitude is reported for post-sitting but no change in P3 amplitude during post-exercise relative to pre-exercise or pre-sitting baseline is reported. None of them report on the N2. This brief overview showcases the lack of consistency in effects and motivates further research regarding the immediate effects. To contribute to our understanding of these immediate effects of an individual acute bout of exercise, the current study developed an approach that allowed ERP measurements 2–3 min after cessation of the exercise intervention and the completion of the cognitive task within 10–11 min after exercising, thereby capturing a narrow time window of the very initial neurocognitive effects post-exercise. There is evidence that, for example, cortisol levels peak about 20–30 min after cessation of an acute bout of moderate-intensity cardiovascular exercise of 15 min (e.g., [45]). In the current study, by restricting experimental testing to a brief time window immediately following exercise (completion within 11 min), the focus is placed on immediate physiological changes and the impact of slower or delayed physiological effects on the obtained results could be reduced.

Also, testing immediately after exercising allows the investigation of neurophysiological effects directly related to exercise without the artefacts or additional effects related to measuring while exercising. Measuring cognition while exercising can be considered a dual-task in which one task might take precedence over the other, which makes it more challenging to filter out the exercise-related effects. We believe that understanding acute effects, including those starting immediately after exercising, contribute to better understanding how long-term effects linked to physical fitness effects on cognition are possibly the result of incremental benefits derived from each acute bout of exercise. Therefore, to understand and possibly explain the long-term effects of physical fitness effects, it is pertinent to investigate the effects at the more “micro”-level of individual bouts of acute exercise, again including those immediately after exercising. One possible explanation for immediate effects that could translate into long-term changes and benefits is an increase in blood flow, which over time yields structural changes such as an increase in vascularization. Other long-term structural changes linked to chronic physical activity, such as the growth in synaptic connections, are likely also driven by repeated short-term changes in neurotransmitter systems (e.g., dopamine) [6].

We hypothesized exercise to have beneficial effects on attentional control. We anticipated shorter response times following moderate-intensity exercise due to the mobilization of attentional control resources [8,10,11,21]. By means of recording ERPs, we further explored potential mechanisms underlying the expected behavioral benefits. We projected that the increase in attentional resources available to meet task demands after exercise would lead to increased P3 ERP amplitudes. Further, we hypothesized shorter P3 latencies as a reflection of faster stimulus evaluation processes [8,16,39,40,43,46]. Given the mixed results regarding the effect of acute exercise on N2 amplitude and latencies and especially due to the paucity of reported immediate effects of acute exercise, no specific hypotheses could be formulated.

## 2. Materials and Methods

### 2.1. Participants

A total of 11 participants (7 women) were recruited and included in the study (mean age 25.64 years; SD = 3.78; range 21–33). Although the sample size is low, the chosen sample size was based on available literature that is comparable with respect to experimental design, sample characteristics, and methodology. There are two studies that closely resemble the current one in that ERPs were recorded during a Flanker task immediately after an acute bout of moderate cardiovascular exercise with a sample size of *N* = 12 [8] and a *n* = 12 subsample comprised of young adults (see supplementary material in [23]). Both studies report significant effects related to post-exercise.

The participants in the current study reported to be healthy and had a normal body mass index (mean BMI = 22.80; SD = 2.28). All subjects had normal or corrected-to-normal vision and had no history of psychiatric or neurological disorders. Participants took part voluntarily and provided written consent. The study was approved by the ethical board of the German Psychological Association (31.08.2012, TM 082012).

### 2.2. Stimuli and Experimental Design (Flanker Task)

Stimuli were presented on a 17-inch iiyama Vision Master 452 CRT monitor set to a resolution of 1024 × 768 pixels. Participants were seated comfortably at a viewing distance of 90 cm from the screen. Stimuli consisted of colored discs (red, green, blue) that took up a visual angle of 0.63° × 0.63° each and were presented on a black background [47]. In the experimental trials, all five discs appeared simultaneously and took up a visual angle of 2.52° × 2.52°. One disc was in the center of the screen, which was flanked by 4 additional discs (one left, one right, one below, and one above; cf. [47]). The space between the central disc and each of the surrounding discs was 0.32°. In between trials, a white fixation cross was presented (0.63° × 0.63°) in the center on the screen. The central disc constituted the target stimulus and was either green or red. The surrounding Flanker discs were either of the same color as the target (i.e., congruent condition (C); all five discs red or green), or of the opposite color, meaning that their color was associated with the opposite response button than the color of the target disc (i.e., incongruent condition (IC); red target and green Flanker or vice versa). This condition induced a response conflict. Stimulus–response mappings were counterbalanced across participants. The third condition was the neutral condition (N) in which the Flanker stimuli were blue and were not associated with a particular response. Thus, the N condition was neutral with respect to the absence of a response conflict as opposed to the IC condition. The neutral condition served as a filler condition and was not of interest in this study and not further considered in the analyses. For a discussion of perceptual conflict (N vs. C), we refer to Winneke and colleagues [47]. The Flanker task was designed to tap into attentional control and the ability to filter out distracting information. In order to avoid potential confounds associated with prepotent response activation and inhibition, which is the case in experimental designs in which the Flanker stimuli appear prior to the central target, the Flanker stimuli and central target in this study appeared at the same time.

The Flanker task consisted of three blocks with 100 trials each. Each trial type (C, IC, N) was equiprobable, and the trial sequence was randomized within each block. Ten second breaks separated the blocks. Each trial consisted of the following series of events (see Figure 1): A central white fixation cross (300 ms), a blank (black) screen (200 ms), stimulus (200 ms), and another blank screen (M = 950 ms; range 800 to 1100 ms). The software Presentation (Neurobehavioral Systems, Albany, Ca, USA) was used for stimulus presentation and behavioral recordings (response time (RT) and accuracy). The RT window was 100 to 1300 ms after stimulus onset. The total duration of the Flanker task was 8.2 min.

### 2.3. Cardiorespiratory Fitness Assessment

Cardiorespiratory fitness data, as indicated by VO_2_max (maximal oxygen consumption), were collected to identify the individual exercise load for the experimental exercise condition. VO_2_max was measured using a computerized indirect calorimetry system (ZAN600, nSpire Health, Oberthulba, Germany) with averages for VO_2_ (oxygen uptake) and respiratory exchange ratio (RER) assessed breath by breath. Participants cycled on an Ergoselect 100 stationary bicycle (ergoline GmbH; Bitz, Germany) at a constant speed and increasing resistance of 15–20 watt/min depending on individual fitness level with electrocardiography activity monitored by a 10-lead fully digital stress system (Kiss, GE Medical Systems) as reported elsewhere [48]. Additionally, participants were fitted with a Polar® heart rate monitor (Polar WearLink+ 31; Polar Electro, Lake Success, NY) to measure heart rate (HR) just before commencement and after completion of the Flanker task. Relative maximum oxygen consumption was expressed in ml/kg/min and was based upon maximal effort as evidenced by an RER ≥ 1.1 [49]. The data from the spiroergometry session (incl. ventilatory, hemodynamic, and metabolic parameters) were analyzed to define the ventilatory anaerobic threshold (VAT) (e.g., [48,49,50,51]). The VAT (sometimes also called aerobic gas exchange threshold ((AerTGE), cf. [52]) was determined by the V-slope method (e.g., [50,52]). In healthy (untrained) individuals, VAT occurs at a VO_2_ value in the range of 40%–60% of VO_2_max (cf. [49,53]). In the current sample, the VAT occurred on average at a VO_2_ value corresponding to 40.90% (SD = 7.41%) of VO_2_max. 

### 2.4. Exercise Intervention

Participants were fitted with a Polar^®^ heart rate monitor (Polar WearLink+ 31; Polar Electro, Lake Success, NY) to measure HR during the exercise intervention and just before commencement and after completion of the Flanker task. Moderate intensity was defined in accordance with a recent meta-analysis on the effect of moderate-intensity exercise on working memory, which used the classification criterion of the heart rate corresponding to 50%–75% of the individual’s VO_2_max ([15]; see also [1]). Furthermore, the largest benefit for complex cognitive tasks (e.g., tasks that demand attentional control) has been associated with moderate-intensity exercise between 40%–80% VO_2_max of a duration of 20 min [2,3]. In this study, the participants’ heart rate was used as intensity index during the exercise intervention. The target heart rate during the exercise intervention corresponded on average to 63.58% (SD = 6.40%) of the HR at VO_2_max. The resistance at the start of the exercise intervention was set to the watt value corresponding to the individual watt value (m = 83.72 watt; SD = 22.83) linked to the target heart rate. This watt value corresponded to 39.84% (SD = 7.78%) of the watt level at VO_2_max. The exercise intervention lasted for 20 min. During the exercise intervention, the critical value of interest was the target HR as index of intensity, which was monitored constantly. If the actual HR deviated from the target HR, the watt value was adjusted until the actual HR was close to the target HR. This adjustment was necessary in four participants: In two cases, the initial watt value had to be increased by 10 watts, in another by 5 watts, and for a fourth participant the watt value had to be lowered by 20 watts. On average, the HR during the exercise intervention differed by only 1.65 beats per minute (BPM; SD = 7.29) from the target HR (see Table 1). In percent, the deviation of the target HR was only 1.66% (SD = 6.63%). Participants reported only light levels of exhaustion confirming that the level of exercise corresponded to moderate intensity both objectively and subjectively. The amount of sweating was, at most, minimal. After the exercise intervention was finished, participants kept on the HR monitor. Their HR at the start of the Flanker task was, on average, 11.21% higher (SD = 11.72%) than their HR at rest (see Table 1). The HR at the end of the Flanker task following exercise was, on average, 3.70 BPM over the resting HR (SD = 7.36 BPM), which amounts to 6.19% (SD = 9.80%).

### 2.5. EEG Data Acquisition and Analyses

A continuous electroencephalogram (EEG) was recorded with 32 AG/AgCl Biosemi ActiveTwo electrodes mounted in an elastic nylon cap (Biosemi, Amsterdam) and arranged according to the international 10/20 system [54]. Vertical and horizontal eye movements were monitored by electro-oculograms (EOG) with electrodes positioned both above and below the left eye and beside the outer canthi of each eye. The EEG signal was recorded at a sampling rate of 2048 Hz in a 0.16 Hz to 100 Hz bandwidth. Subsequent data processing steps were performed using Brain Vision Analyzer 2 software (Brain Products GmbH, Gilching, Germany). All EEG data were down-sampled offline to 512 Hz and re-referenced offline to linked mastoids. The continuous EEG was divided into 700 ms epochs with time point 0 as the stimulus onset and a 100 ms prestimulus baseline interval, which was used to baseline correct EEG waveforms. The EEG was filtered offline for frequencies between 0.1 and 40 Hz and by use of a 50 Hz notch filter. Trials with EOG activity were corrected using the Brain Vision Analyzer built-in Gratton and Coles ocular correction function. Trials containing EEG activity exceeding ±75 µV were rejected. Only trials with correct responses in the Flanker task were included in ERP analyses (on average 69.64 (SD = 17.85; min = 41) congruent trials and 67.14 (SD = 15.93; min = 41) incongruent trials). Four measures were extracted from the ERP waveforms, namely, the peak amplitude (measured in µV) and the peak latency (measured in ms) of the N2 and the P3. On the basis of the grand average, the N2 was defined as the most negative peak between 190 and 320 ms after stimulus onset at electrode site Fz and the P3 was defined as the most positive peak between 260 and 470 ms after stimulus onset at electrode site Pz. A semiautomatic peak detection function of the Brain Vision Analyzer 2 software was applied to averaged waveforms of each individual, followed by manual check and adjustment if necessary.

### 2.6. Testing Procedure

Participants were invited to three sessions. They were asked to get sufficient sleep the night before and come in well rested, to refrain from consuming alcohol and drugs the night before, as well as to limit their caffeine intake to one cup of tea/coffee in the morning. Importantly, they were asked not to exercise prior to coming to the lab. In their first session, all participants completed a consent form before participants’ height and weight were measured. Afterwards, the cardiorespiratory fitness assessment was conducted. The other two sessions entailed the exercise intervention at moderate intensity followed by the Flanker task (Exercise-condition) or the Flanker task at rest without any exercise during the session (Rest-condition). The sequence of the two test sessions was counterbalanced to control for potential test-retest effects. Half of the participants did the Rest-condition in the second testing session and the Exercise-condition in the third session (*n* = 5). The remaining six participants performed the Exercise-condition in the second session and in the third testing session the Rest-condition. After arriving in the laboratory, participants were asked to sit for a few min to ensure that HR returned to resting pulse and to get acquainted with the testing environment. Then, the EEG setup procedure was described, and participants were fitted with an EEG cap and EOG electrodes and were seated comfortably in a sound-attenuated room where the EEG testing took place. After the cap was prepared, the experiment was explained, and any open questions were answered. Therefore, before the task commenced, participants were resting in a seated position long enough to exclude any exercise-related arousal before coming to the lab, which could confound the results of the rest-condition. After the preparation phase and after the task was explained, 20 practice trials of the Flanker task were administered to ensure participants understood the task. In the Rest-condition, the Flanker task started after the practice trials were completed. In the Exercise-condition, participants were fitted with a Polar^®^ heart rate monitor, and after completion of the Flanker practice trials, they were asked to mount a stationary bicycle and start the exercise intervention at moderate intensity. Importantly, participants remained fully equipped with the EEG cap and EOG electrodes during the exercise to ensure that the Flanker task post-exercise could commence immediately after the 20-minute exercise intervention. The average time between termination of the exercise intervention and the start of the post-exercise Flanker task was 2.56 min (SD = 0.40 range: 2.00–3.10 min). To ensure good EEG data quality, the impedance values were checked before the Flanker task started. After Flanker task completion, the EEG cap and EOG electrodes were removed.

### 2.7. Statistics

2 × 2 repeated measures ANOVAs (2 (congruence: Congruent vs. incongruent) × 2 (test time: Rest vs. after exercise) separately for the dependent variables response time in ms (RT) and accuracy defined as percent correct responses were performed. ERP data (N2 peak amplitude (µV) and peak latency (ms); P3 peak amplitude (µV) and peak latency (ms)) were analyzed separately using a 2 × 2 × 6 ANOVA with the factors congruence, test time, and electrode site, separately for N2 and P3. All repeated-measures ANOVAs were adjusted with the Greenhouse-Geisser non-sphericity correction [55] for effects with more than one degree of freedom (df) in the numerator. According to convention, uncorrected degrees of freedom, mean square error (MSE), partial eta-square (η^2^), and adjusted p values are reported. Significant main effects and interactions were followed by Bonferroni corrected analyses of simple effects and, unless stated otherwise, the differences reported are significant at α = 0.05 or below.

## 3. Results

### 3.1. Behavioral Data

The behavioral results are depicted in Figure 2 and statistical results are summarized in Table 2. Analysis of response times revealed a main effect of congruence due to faster responses in the congruent compared to incongruent trials (Figure 2a). No main effect of test time or an interaction effect was found for response time data. Similarly, analyses of response accuracy showed a main effect of congruence with more accurate responses in the congruent than in the incongruent condition (Figure 2b). Neither a main effect of test time nor an interaction effect was detected for accuracy data.

### 3.2. ERP Data: N2

Visual inspection of the waveforms as well as topographical plots (Figure 3) indicate that N2 is most prominent at frontal sites. N2 was largest at Fz confirmed by descriptive analysis using SPSS based on extracted peak data for incongruent and congruent conditions both at rest and after exercise. Separate 2 × 2 × 6 repeated measures ANOVAs with factors congruence (congruent and incongruent), test time (at rest and after acute exercise) and electrode site (Cz, FC1, FC2, F3, F4, Fz) were conducted for N2 latency and amplitude. The ERP waveforms are depicted in Figure 3 and N2 peak values of Fz are shown in Table 3 and Figure 4 visualizes group mean and individual mean N2 peak amplitudes and latencies across electrode sites Cz, FC1, FC2, F3, F4, and Fz. Statistical results are summarized in Table 4.

The analysis of N2 latency values revealed a main effect of electrode site due to earlier N2 latencies at frontal sites F3, F4, and Fz compared to more central sites FC1, FC2, and Cz.

The ANOVA of N2 amplitude showed a main effect of congruence due to larger (i.e., more negative) N2 peak amplitude values in the incongruent condition (M = −2.10; SD = 2.08) than in the congruent condition (M = −1.09; SD = 2.45). This effect was superseded by a congruence × site interaction due to more pronounced incongruency effects at frontal sites F3, F4, and Fz relative to central sites. No statistically significant main effect of test time or an interaction effect involving the factor test time was found for N2 amplitude data.

To further explore the temporal aspect of the effect of exercise on the N2, the dataset was split in half to compare the effects of the first and the second half. The N2 data were reanalyzed using a 2 × 2 × 2 repeated measures ANOVA with the factors test time (rest vs. exercise), congruence (congruent vs. incongruent), and half (first vs. second half). To reduce the complexity of the ANOVA, we eliminated the factor electrode site and we used the data from the average of the fronto-central electrode sites (Cz, FC1, FC2, F3, F4, Fz). With respect to latency, no significant effects were detected. For N2 amplitude, the main effect of congruence (i.e., larger N2 amplitude for incongruent trials) was confirmed (F(1,10) = 8.91, MSE = 3.38, partial η^2^ = 0.471, *p* = 0.014). Furthermore, a significant test time × half interaction was found (F(1,10) = 5.67, MSE = 1.13, partial η^2^ = 0.362, *p* = 0.039). This interaction is due to larger N2 amplitude values in the second half of the data after exercising than in the second half at rest. As can be seen in Figure 5, the amplitude difference in the second half between rest and exercise is due to a reduction in N2 amplitude after rest, whereas N2 amplitude is maintained or even slightly increases in the second half of trials after exercising.

### 3.3. ERP Data: P3

Visual inspection of the waveforms as well as topographical plots (Figure 6) indicates that P3 is most prominent at parietal sites. P3 is largest at Pz confirmed by descriptive analysis using SPSS based on extracted peak data for incongruent and congruent conditions both at rest and after exercise. Separate 2 × 2 × 6 repeated measures ANOVAs with factors congruence (congruent and incongruent), test time (at rest and after acute exercise), and electrode site (Cz, CP1, CP2, P3, P4, Pz) were conducted for latency data and amplitude for the P3 component. The ERP waveforms are depicted in Figure 6, P3 peak values of Pz are shown in Table 5, and Figure 7 visualizes group mean and individual mean P3 peak amplitudes and latencies across electrode sites Cz, CP1, CP2, P3, P4, and Pz. Statistical results are summarized in Table 4.

The analysis of P3 latency values showed a main effect of test time driven by significantly earlier P3 peak latencies after exercise (M = 350.31 ms; SD = 14.92) than at rest (M = 362.62 ms; SD = 20.73) as well as a main effect of congruence due to earlier P3 peak latencies in the easier congruent condition (M = 348.74; SD = 15.38) than in the more difficult incongruent condition (M = 364.18; SD = 21.99). A main effect of electrode site is due to earlier P3 latencies at parietal sites P3, P4, and Pz (M = 346.74; SD = 18.10) compared to more central CP1, CP2, and Cz (M = 366.18; SD = 26.35). No statistically significant interaction effects became apparent. Note that due to non-linearity of peak scoring measures, statistical effects are not necessarily reflected in the depiction of grand average waveforms [56].

The analysis of P3 amplitude values showed a main effect of congruence due to larger P3 peak amplitude values in the congruent condition (M = 9.27; SD = 4.03) than in the incongruent condition (M = 7.13; SD = 2.90). No statistically significant main effect of test time or an interaction effect was found.

The split half analysis of P3 latency confirmed the above-reported main effect of congruence (F(1,10) = 10.45, MSE = 338.28, partial η^2^ = 0.511, *p* = 0.009) due to larger P3 amplitudes for congruent than incongruent stimuli as well as the main effect of test time (F(1,10) = 7.63, MSE = 596.15, partial η^2^ = 0.433, *p* = 0.02) with earlier P3 peak latencies after exercise than at rest.

For P3 amplitude, the main effect of congruence was confirmed (F(1,10) = 21.50, MSE = 1.88, partial η^2^ = 0.683, *p* = 0.001) but superseded by a congruence × half interaction (F(1,10) = 5.85, MSE = 0.93, partial η^2^ = 0.370, *p* = 0.036) due to smaller P3 amplitudes to incongruent trials in the second half as compared to the first half (see Figure 8).

## 4. Discussion

In this study, we investigated the immediate effects of an acute bout of exercise of moderate intensity on attentional control as assessed by a Flanker task. Of interest were both behavioral as well as neurophysiological indices. The P3 peak occurred earlier after exercise as compared to rest, indicating faster neural processing. No other significant neurophysiological or behavioral effects of acute exercise were found.

As expected, response time and accuracy data revealed the distracting effect of the incongruent Flanker stimuli [24]. This can be seen in a higher error rate and slower response times in the incongruent as compared to the congruent condition. Similarly, the N2 data showed the expected interference effect reflected in more negative N2 amplitude values in incongruent than congruent trials. Data regarding the P3 are also in line with the expected effects of distracting Flanker stimuli (e.g., [16]). For incongruent trials, the P3 peak was delayed and decreased in amplitude relative to the congruent condition. This shows that the manipulation of the Flanker task with respect to inducing processing costs (both behavioral and neural) as a result of the inhibition of the interfering influence of distracting stimuli was successful.

However, neither behavioral nor N2 ERP data revealed statistically significant effects of exercise when analyzing the data for the entire duration of the flanker task. In the literature, results regarding the effect of exercise on the N2 are mixed [11,19,37], whereas for behavioral data, some studies report faster response times following exercise (e.g., [8,18]). A possible explanation for the absence of behavioral and N2 effects is that the task was not challenging enough and that in order to reveal effects, possibly more difficult tasks have to be used [20]. This might also apply to the results of the P3 amplitude. Contrary to the hypothesis, the data do not reveal an increase in P3 amplitude following exercise. Such an increase would reflect an increase in resources to inhibit neuronal activity unrelated to task demands and facilitate attentional processing [20]. Pontifex and colleagues [20], using an oddball task that requires ignoring frequent stimuli or distractors and selectively responding and attending to a less frequent target stimulus, also did not find an increase in the P3 component following a 20-minute bout of moderate exercise relative to a pre-exercise or a pre-sitting baseline. As stated earlier, a marked difference in P3 amplitude between post-sitting vs. post-exercising was reported, driven by a significant decrease in P3 amplitude for post-sitting measurements indicating that exercise helps maintain attentional resources. Their study resembles the current one with respect to sample characteristics, namely high-functioning university students, as well as time of testing post-exercise. Pontifex and colleagues [20] started the experimental session 4.5 min following the exercise bout. Post-exercise testing in the current study commenced even earlier (2–3 min) and the entire task was completed in 8 min. In addition to Pontifex et al. [20] and the current study, only a limited number of studies commenced experimental EEG recording immediately after (i.e., within 5 min) exercise but employ a variety of cognitive tasks [8,21,22,43]. The majority of studies on acute effects of exercise on cognition and ERPs conduct experimental cognitive tasks with some delay (e.g., 48 min [16], 20 min [17,18], 15 min [19], 16 min [14]). The discrepancy in reported effects of exercise on P3 amplitude suggests that time of testing is critical and the immediate (i.e., test completion within 10 min of exercise cessation) effects might be less robust than those at delayed post-exercise measurement points. Related to this, another critical aspect to consider is the possibility that a task, depending on duration, stretches over multiple post-exercise periods and, hence, performance over these intervals might be averaged and differences diluted (see [7]). To check for time interval effects, we conducted a post-hoc split half analysis on behavioral data and compared performance of the first and second half of the flanker task (i.e., 0–4 min and 4–8 min) both at rest and after exercise. However, no statistically significant effect of interval was apparent, suggesting that with respect to behavioral parameters, the chosen task duration was brief enough to likely capture the immediate effects of the first post-exercise period. A split half analysis of P3 amplitude did not reveal a differential effect of exercise during the first or second half of the flanker task. However, the split half analysis of the N2 amplitude data revealed an interesting half × test time interaction. Whereas N2 amplitude decreased during the second half of the Flanker trials after rest (i.e., became less negative), N2 amplitude after exercising remained constant or even increased (i.e., became more negative) during the second half. This suggests that exercise contributes to maintenance of neurocognitive resources to possibly counter a loss in concentration or attentional processing already within the first 5 min after exercising. This pattern is similar to the above-mentioned findings regarding P3 amplitude by Pontifex et al. (2015) who report a decrease in P3 amplitude after sitting but maintenance of P3 amplitude after exercising.

Regarding P3 ERP latency, the reported data support our hypothesis of faster neural processing following acute exercise of moderate intensity. This is in line with previous research that also reports earlier P3 peaks after an acute bout of exercise (e.g., [8,16,39,40,46]). Shorter P3 peak latencies can be interpreted as an indication of faster stimulus evaluation processes [34,35,36]. This interpretation is supported by the finding of a delayed P3 peak in incongruent as compared to congruent trials (Table 5). We also conducted a post-hoc split half analysis of P3 latency, which revealed that faster P3 latencies were independent of the factor half. This is interesting with respect to N2 amplitude analysis, which revealed an effect of half. Whereas the P3 component is modulated immediately after exercise and throughout the entire Flanker task, the effects of exercise on the N2 component start to emerge with a slight delay (i.e., as of the second half). In other words, time of assessment is important as it allows constructing a timeline of effects and processes linked to post-exercise effects.

One question arises as to the finding of latency shifts in the P3 but not for response times. It is known that a physical response to a stimulus such as a button press involves at least two processes, stimulus evaluation and response selection and execution. The latency of the P3 reflects the timing of the stimulus evaluation process. Previous research has shown that the timing of the stimulus evaluation process is independent of the timing of response selection particularly under speeded responses conditions [57]. Although, stimulus evaluation precedes response selection, it is not unlikely that in tasks where participants try to respond fast, a response is initiated before stimulus evaluation is completed [57]. In other words, the processes do not always have to follow sequentially but can partly overlap. This is a possible explanation for the lack of behavioral response time effects due to exercise, despite the fact that P3 latency showed a significant time shift indicative of faster processing. Hillman and colleagues [16] reported a significant P3 latency shift following exercise relative to rest but no behavioral effects. Studies using the Stroop task [19,38] report faster response times after acute exercise but do not find P3 peak latency shifts. Taken together, the results indicate that P3 latency, as an index of stimulus evaluation processes, is not always tightly coupled with behavioral response times. As mentioned above, it is also possible that the task was too easy and that in order to highlight a behavioral benefit, a more cognitively demanding task might be necessary [20].

Given that testing commenced within 3 min of exercising, findings regarding effects on cognition while exercising seem relevant to be briefly discussed. Despite the temporal proximity and therefore the potential of comparable effects of exercise on underlying processes, the nature of the tasks bears important differences. A cognitive task while exercising (treadmill or ergometer) naturally has dual-task characteristics whereas performing a cognitive task after exercising does not. A study using a flanker task during moderate exercise reports an exercise-related P3 amplitude increase but only fronto-laterally [37]. Given the topography, this likely corresponds to the P3a subcomponent [28] and not to the parietal P3b component related to stimulus evaluation processes and commonly reported in studies on acute exercise effects. Furthermore, larger N2 amplitudes were also reported at fronto-lateral sites. Although the split half analysis of the N2 data of the current study did not reveal larger N2 amplitudes after exercising, the results seem to stipulate that exercise can counter a reduction of attentional resources devoted to conflict resolution as seen in the rest condition. Regarding latencies, the authors report a slowing of both N2 and P3 during exercise. P3 latency slowing is also shown by a long-duration exercise study by Grego and colleagues [58]. Based on hormonal changes (epinephrine and cortisol) measured in parallel, the authors interpret the effects while exercising in terms of increased arousal, as P3 latencies peak earlier after exercise compared to P3s while exercising. However, other studies recording the P3 while exercising report earlier P3 latencies (e.g., [59]). As for studies commencing after an acute bout of exercise, these studies differ in terms of chosen task, duration of exercise, and with respect to the intensity of exercise. Accordingly, results vary, but it appears that results from studies conducting neurocognitive tasks during exercise should be separated from studies immediately following exercise as they seem to fall in different (post-) exercise intervals with respect to effect on performance possibly due to differences in arousal [7], shared neural resources as in dual task effects [3], or different levels of exercise-induced neural noise [15]. Although the current study does not disentangle the effects of exercise on cognition while exercising as compared to immediately after, it supports the notion that processes of stimulus evaluation are faster immediately after an acute bout of exercise.

The underlying mechanisms concerning the beneficial effects of exercise on cognition are not clearly understood. One prominent theory proposes that exercise leads to increases in arousal, which activates attentional and memory resources yielding benefits for executive functioning (e.g., [5,15,60]). With respect to the role of exercise as an enhancing agent for cognition, research has shown that exercise increases availability of catecholamines in the central nervous system of rats [61], in particular dopamine (DA) [62]. In humans, elevated DA plasma levels have been associated with acute physical exercise [63,64]. Furthermore, DA is central to executive processing in the brain [65,66]. The results from this study strengthen interpretations from previous research regarding underlying mechanisms of exercise effects on neurocognitive functioning. Electrophysiological responses are the result of electrochemical actions of neurotransmitter systems (e.g., DA), while participants perform cognitive tasks. The P3 component related to stimulus evaluation processes are most likely generated by cortical cells in temporo-parietal areas and involves the dopaminergic neurotransmitter system [67]. It has been shown that the striatal DA system influences P3 amplitude and latency [68]. Therefore, acute exercise might modify the activity of the DA system and thereby enhance neurophysiological processes. In the course of repeated exercise, each bout of exercise facilitates long-term potentiation processes. These, in turn, can manifest themselves in neurostructural changes as well as cognitive benefits associated with chronic exercise [6,46]. Although, the results of the present study could be integrated in a framework involving dopamine as a central component to explain the effect of acute exercise on cognition and electrophysiology, they do not reflect direct evidence and further research is needed to tease apart the direct associations.

### Limitations

Apart from the above-mentioned experimental design aspects regarding the focus on early post-exercise effects and not extending to late effects as well, the limitation of small sample size should be taken into account as explanation for discrepant findings compared to other studies. Even though the chosen sample size is comparable to other studies using very similar experimental approaches and report significant effects (e.g., [8,21,69]), the sample size is low and therefore might not detect more subtle effects. Likewise, one issue with small sample sizes in general is the detection of true effects, and more studies also involving different populations are necessary to confirm the reported effects. A very current review provides a nice overview of experimental issues, including sample size, which should be taken into consideration when designing a study investigating the effects of acute exercise on cognition [23]. Another limitation is that no heart rate recordings were obtained for the rest condition. Baseline HR at rest was only obtained before the exercise condition (see Table 1). Although unlikely, as HR in young adults under similar conditions is stable, the possibility of systematic differences in HR between the rest measurement and after exercise measurement cannot be completely ruled out. Even though attentional control is an important cognitive skill linked to executive functioning, other cognitive abilities such as (working) memory, decision making, or perceptual skills are of importance as well but were not investigated in this study. Therefore, more research is necessary to understand the immediate effects of exercise on other cognitive skills to obtain a more comprehensive picture also with respect to the various underlying neurophysiological processes and how and when they are modulated by exercise. Also, this study focused on the effects of exercise on cognition and contrasted this to recordings at rest rather than at pre- vs. post-exercise comparisons. Regarding the rest condition, it should be noted that it did not entail a standardized activity such as watching a movie. The focus was placed on ensuring that the rest condition entailed a sufficiently long time period without any physical exercise to control for any exercise-related arousal in order to contrast the effect of acute exercise to a non-exercise control condition. Finally, the behavioral results indicate that performance was very high. As mentioned above, the chosen task was possibly too easy for the sample of this study to reveal behavioral effects of exercise. Future studies should account for that and administer tasks that are challenging enough for the study sample to allow for behavioral effects to transpire.

Like previous research, this study shows individual differences in the responses to acute exercise and its impact on cognition. Different factors might contribute to individual differences. Although all participants were moderately fit, their base fitness levels still varied. Even though the exercise levels were individually adjusted based on the VO2max levels, it is possible that fitness level plays a role even when differences are not extreme as in sedentary vs. high-fitness individuals. Although not investigated in this study, the sports participants are engaged in could play a role when using a task of attentional control and the ability to inhibit distracting information. Some sports such as team ball sports might hone attentional control skills to a larger extent than running or swimming. Therefore, more research is needed to better understand the impact of these factors on attentional control and exercise. Needless to say, physical activity is recommended for anyone, but naturally there are certain groups in the population that might extra benefits. These include adults who show age-related cognitive decline and individuals with cognitive impairments due to psychiatric or psychological disorders or traumatic brain damage. Particularly safety critical professions such as air traffic control or jobs in control centers place a high demand on attentional control. Future applied research projects could address the question whether brief bouts of acute exercise can improve attentional control in real-world applications. In this context, future studies are necessary to investigate more closely how long the beneficial effects of acute bouts of exercise persist. That information is relevant, for example, in order to effectively setup schedules at school or at work and include, as far as possible, active breaks.

## 5. Conclusions

The results of the current study add to existing research to further validate and confirm findings on the enhancing effects of acute exercise on neurocognitive processes. Furthermore, the results of the EEG data contribute to our knowledge and understanding of the potential underlying neural mechanisms involved in exercise-related benefits to cognitive functioning immediately following exercise (within 5 min). Given that some neurobiological processes respond faster than others, it is possible that some processes are modulated after a slight delay (e.g., hormonal changes) compared to processes that respond faster (e.g., blood flow). Only by measuring very early it is possible to also detect immediate effects. The split half analysis of the short flanker task administered in this study indicates that the time point of measuring post exercise can make a difference. Whereas the effect of exercise on the P3 component (P3 latency) was apparent throughout the entire duration of the flanker task, an effect of exercise on the N2 component (N2 amplitude) became apparent only in the second half of the flanker task. In other words, the underlying processes of the N2 and the P3 seem to be affected differently by exercise.

To understand cognitive benefits associated with physical fitness as a result of long-term regular chronic exercise, we believe it is essential to also elucidate the immediate, short-term effects of physical activity. This is to say, each individual acute bout of exercise, if repeated on a regular basis as in chronic exercise, should, over time, manifest itself in more permanent changes yielding long-term cognitive benefits both at the behavioral as well as at a neuro-structural level (e.g., [6]).

This study confirms that it is possible to evaluate the immediate impact (2–3 min after exercising) of a moderate acute bout of exercise on attentional control, which is a cognitive skill that is relevant to many everyday cognitive demands. This could be of particular importance for companies that are interested in promoting physical activity at the workplace. One implementation could encourage individuals to (partly) walk or bike to the workplace and/or to include a brief but brisk walk during lunch break. However, it is still an open research question for future studies for how long the exercise-related effects on cognitive functions last.

## Figures and Tables

**Figure 1 jcm-08-01348-f001:**
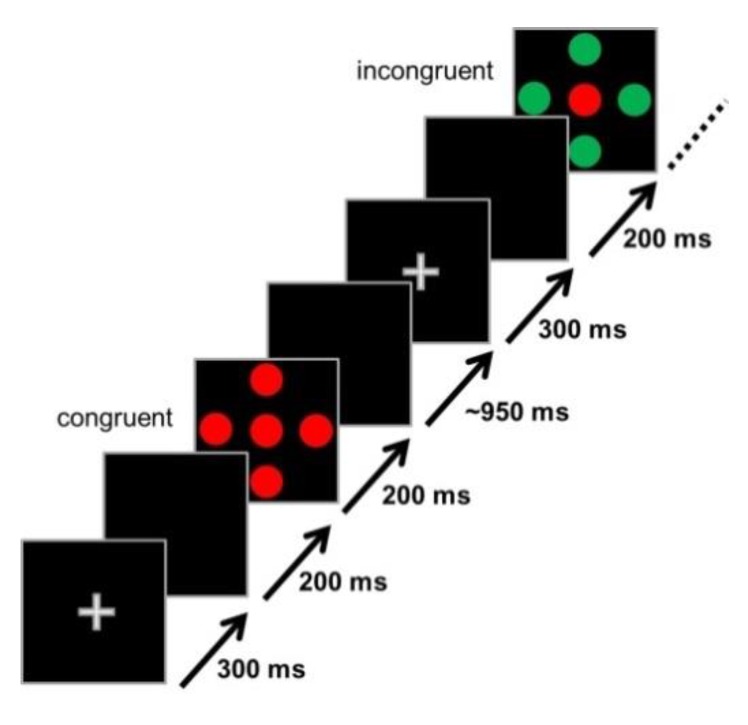
Schematic depiction of the Flanker task used in this study showing an example of a congruent and an incongruent trial.

**Figure 2 jcm-08-01348-f002:**
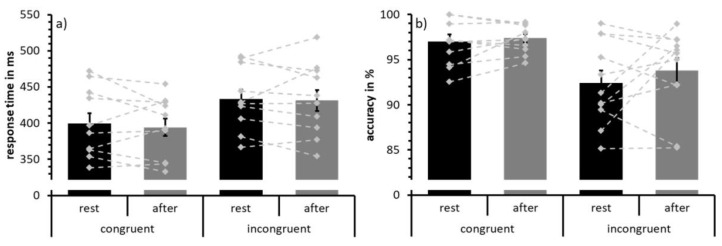
Mean response times (**a**) and response accuracy (**b**) to congruent and incongruent trials at rest (black) and following acute exercise at moderate intensity (grey). Error bars depict standard error values. The DotPlot visualizes the individual data (light grey) for the congruent and incongruent condition as well as the change from rest to after-exercise (dashed lines).

**Figure 3 jcm-08-01348-f003:**
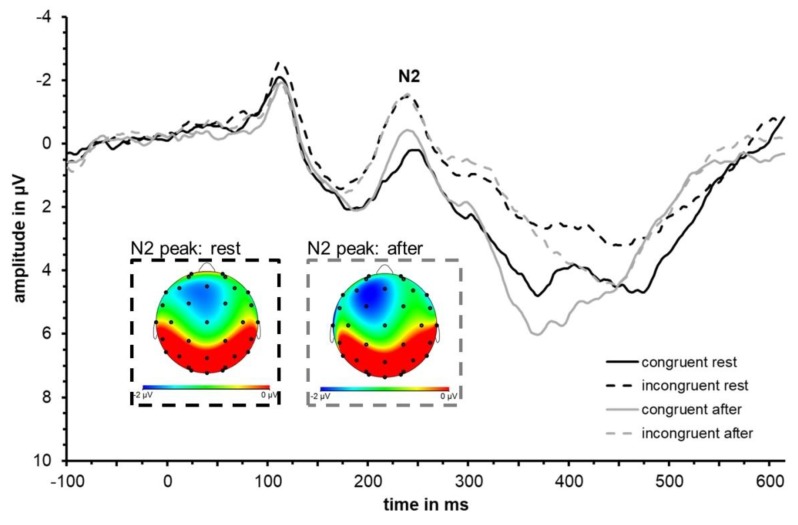
Grand average event-related potential (ERP) waveform (*N* = 11) averaged across fronto-central electrode sites (F3, Fz, F4, FC1, FC2, Cz) depicting the neural responses (in particular the N2) to congruent (solid) and incongruent (dashed) trials at rest (black) and immediately after an acute bout of moderate-intensity exercise (grey). Topographical distributions at the time of N2 peak are shown for the incongruent condition at rest (dashed, black frame) and after exercise (dashed, grey frame).

**Figure 4 jcm-08-01348-f004:**
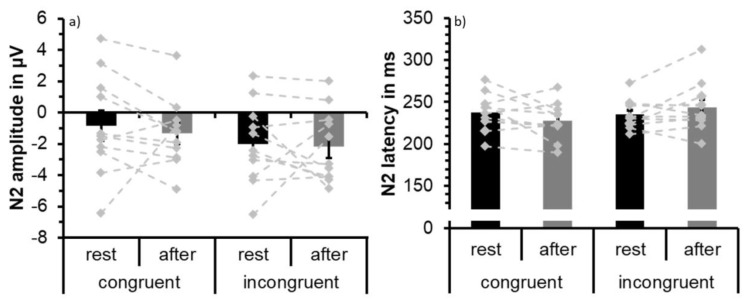
Mean N2 peak amplitude (**a**) and peak latency (**b**) averaged across participants (*N* = 11) and across fronto-central electrode sites (F3, Fz, F4, FC1, FC2, Cz) to congruent and incongruent trials at rest (black) and following acute exercise at moderate intensity (grey). Error bars depict standard error values. The DotPlot visualizes the individual data (light grey) for the congruent and incongruent condition as well as the change from rest to after-exercise (dashed lines).

**Figure 5 jcm-08-01348-f005:**
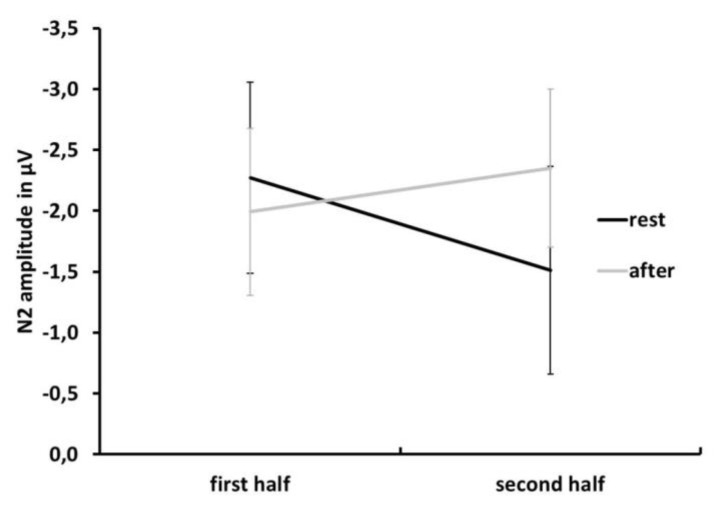
Mean N2 peak amplitude at rest (black) and after exercise (grey) trials in the first and second half of the dataset to visualize the test time × half interaction. Note: Negativity is plotted upward. Error bars depict standard error values.

**Figure 6 jcm-08-01348-f006:**
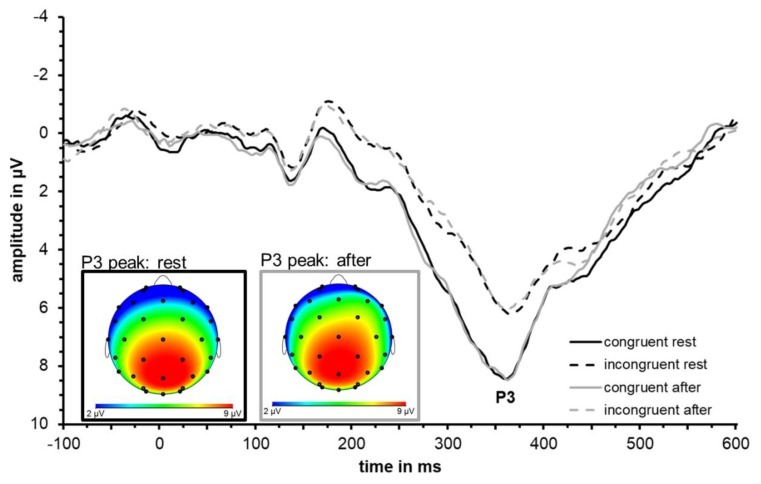
Grand average ERP waveform (*N* = 11) averaged across parieto-central electrode sites (P3, Pz, P4, CP1, CP2, Cz) depicting the neural responses (in particular the P3) to congruent (solid) and incongruent (dashed) trials at rest (black) and immediately after an acute bout of moderate-intensity exercise (grey). Topographical distributions at the time of P3 peak are shown for the congruent condition at rest (solid, black frame) and after exercise (solid, grey frame).

**Figure 7 jcm-08-01348-f007:**
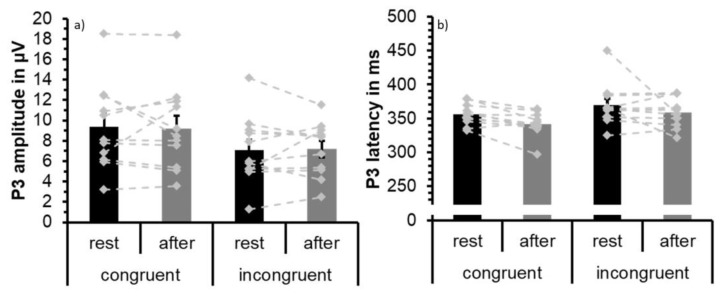
Mean P3 peak amplitude (**a**) and peak latency (**b**) averaged across participants (*N* = 11) and across parieto-central electrode sites (P3, Pz, P4, CP1, CP2, Cz) to congruent and incongruent trials at rest (black) and following acute exercise at moderate intensity (grey). Error bars depict standard error values. The DotPlot visualizes the individual data (light grey) for the congruent and incongruent condition as well as the change from rest to after-exercise (dashed lines).

**Figure 8 jcm-08-01348-f008:**
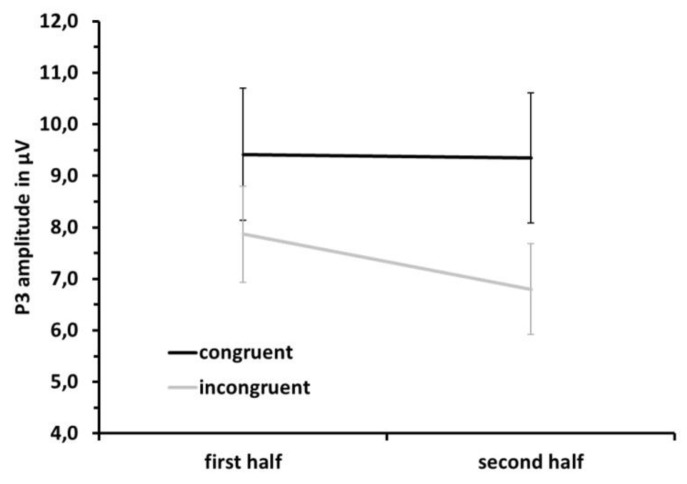
Mean P3 peak amplitude for congruent (black) and incongruent (grey) trials in the first and second half of the dataset to visualize the congruence × half interaction. Error bars depict standard error values.

**Table 1 jcm-08-01348-t001:** Average heart rate values (beats per minute (BPM)) at rest, during the intervention (minutes 2 to 20), after 20 min of exercising, at the start and end of the Flanker task, as well as the target heart rate.

BPM	Target	At Rest	Avg. during Exercise	After 20 min. Exercise	Flanker Start	Flanker End
Avg.	113.00	74.73	114.65	119.55	82.36	78.43
SD	13.21	12.92	12.98	14.25	11.50	9.13
Min	96.00	60.00	92.40	95.00	68.00	68.00
Max	140.00	100.00	139.20	149.00	100.00	97.00

**Table 2 jcm-08-01348-t002:** Statistics on the analyses of response times and response accuracy.

**Response time**	**F(df)**	**MSE**	**partial η^2^**	***p***
Condition	F(1,10) = 52.42	274.84	0.840	<0.001 *
Time	F(1,10) = 0.03	520.72	0.003	0.862
Condition × Time	F(1,10) = 0.003	217.06	0.000	0.960
**Response accuracy**	**F(df)**	**MSE**	**partial η^2^**	***p***
Condition	F(1,10) = 16.76	11.09	0.626	0.002 *
Time	F(1,10) = 1.72	4.81	0.147	0.219
Condition × Time	F(1,10) = 0.42	6.49	0.041	0.530

* *p* < 0.05.

**Table 3 jcm-08-01348-t003:** Mean (SD) N2 peak latencies and amplitude for congruent and incongruent conditions at rest and after exercise at electrode site Fz.

N2	Rest		After Exercise	
	Latency in ms	Amplitude in µV	Latency in ms	Amplitude in µV
Congruent	238.81 (28.56)	−1.07 (3.75)	231.00 (27.57)	−1.40 (2.58)
Incongruent	243.79 (30.76)	−2.36 (3.03)	245.03 (30.39)	−2.55 (2.66)

**Table 4 jcm-08-01348-t004:** Statistics on the analyses of amplitude and latency of the N2 and P3.

N2 Amplitude	F(DF)	MSE	Partial H^2^	*P*
Condition	F(1,10) = 7.25	9.41	0.420	0.023 *
Time	F(1,10) = 0.25	30.71	0.024	0.628
Electrode	F(5,50) = 1.14	21.75	0.102	0.328
Cond × Time	F(1,10) = 0.24	7.46	0.024	0.634
Cond × Elec	F(5,50) = 3.33	1.00	0.25	0.011 *
Time × Elec	F(5,50) = 0.50	2.42	0.047	0.604
Cond × Time × Elec	F(5,50) = 0.93	0.68	0.085	0.422
**N2 Latency**				
Condition	F(1,10) = 0.85	3733.30	0.078	0.379
Time	F(1,10) = 0.02	1372.17	0.002	0.884
Electrode	F(5,50) = 3.26	328.41	0.246	0.044 *
Cond × Time	F(1,10) = 4.89	1179.93	0.328	0.051
Cond × Elec	F(5,50) = 1.25	265.30	0.11	0.310
Time × Elec	F(5,50) = 0.36	398.45	0.035	0.765
Cond × Time × Elec	F(5,50) = 1.42	193.25	0.125	0.259
**P3 Amplitude**				
Condition	F(1,10) = 25.98	11.59	0.722	<0.001 *
Time	F(1,10) = 0.002	21.21	0.000	0.968
Electrode	F(5,50) = 0.99	17.98	0.090	0.384
Cond × Time	F(1,10) = 0.37	3.75	0.036	0.554
Cond × Elec	F(5,50) = 0.91	2.06	0.084	0.427
Time × Elec	F(5,50) = 0.40	3.86	0.038	0.635
Cond × Time × Elec	F(5,50) = 1.75	0.82	0.149	0.185
**P3 Latency**				
Condition	F(1,10) = 6.34	2481.69	0.388	0.030 *
Time	F(1,10) = 5.99	1670.05	0.375	0.034 *
Electrode	F(5,50) = 4.46	4965.29	0.309	0.049 *
Cond × Time	F(1,10) =0.09	1864.47	0.009	0.774
Cond × Elec	F(5,50) = 0.72	210.94	0.067	0.561
Time × Elec	F(5,50) = 0.93	718.05	0.085	0.416
Cond × Time × Elec	F(5,50) = 0.48	653.62	0.046	0.646

* *p* < 0.05.

**Table 5 jcm-08-01348-t005:** Mean (SD) P3 peak latencies and amplitude for congruent and incongruent conditions at rest and after exercise at electrode site Pz.

P3.	Rest		After Exercise	
	Latency in ms	Amplitude in µV	Latency in ms	Amplitude in µV
Congruent	346.59 (20.49)	10.11 (4.36)	336.47 (16.87)	9.76 (4.06)
Incongruent	364.17 (32.41)	7.91 (3.54)	350.32 (18.96)	7.49 (2.62)

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
