# Peer review of "Moderate Cardiovascular Exercise Speeds Up Neural Markers of Stimulus Evaluation During Attentional Control Processes"

_jcm, 2019, doi:10.3390/jcm8091348_

Round 1

Reviewer 1 Report

The revision has addressed my concerns well.

I think some further improvement can be made in the manuscript. To strengthen the significance and broader impact of the research, the discussion could briefly touch on subject characteristics and individual differences in terms of their abilities to do the task and how future studies could be conducted to probe beneficial effects of acute physical exercise in relation to inhibitory control,  aging,  and clinical populations for whom attentional control processes are critical.

Reviewer 2 Report

The revisions do not address reviewer’s initial comments (no response to reviewer’s initial comments has been included). The major methodological flaws including the lack of a control condition and a very small sample size prevent the reviewer from recommending the manuscript for publication in JCM.

The reviewer has further concerns regarding the novelty and significance of the study, which have not been satisfactorily addressed.

1.     The study does not add new information to the extant literature as the effects of acute moderate intensity exercise on behavioral (using a modified flanker task) and neuroelectric indices (P3 component of the ERPs) have been previously reported in the population of young adults (including the effects measured within the first 10 minutes following exercise). The effects of the acute aerobic exercise on N2 component have also been previously reported.

2. The significance of the study is poorly justified (please refer to the initial review for details).

3.     Lastly, the issue with a small sample size not only pertains to the power of the study to detect small effects, but a small sample size raises concerns regarding the reliability of the reported findings.

Reviewer 3 Report

The reviewer would like to commend for the authors’ time and effort on the revision.

The authors attempted to justify the significance of research on cognition immediately following exercise in the responses to the reviewer’s first comment but the arguments provided in their responses are still not very clear. For example, the authors mentioned that investigating the immediate exercise effects “…could help better understand how long-term effects linked to physical fitness effects on cognition…”, but how the understanding of these immediate effects would help better understand chronic exercise effects on cognition? The authors also argued that investigating immediately aftereffects of exercise makes it possible to examine “neurophysiological effects directly related to exercise without artefacts of additional effects related to measuring while exercise”. Does this imply that the authors are actually more interested in cognition during exercise? Regardless, the justifications provided should be incorporated into the introduction of the manuscript rather than just in the responses to reviewers.

The reviewer still has difficulty following some of the statements in the manuscript. For example, Pontifex et al. (2015) did not find slowing of P3 latency following exercise. If the authors read their paper carefully, they found a main effect of time, with longer P3b latency at posttest relative to pretest. This means the P3 latency increased from pretest to posttest regardless of exercise or control conditions. This is different from the authors statement “Pontifex et al. [20] observe no effect on amplitude but a slowing of P3 latencies when comparing before to after exercising” because the decrease in P3 latency was not a result of exercise. This is not a major issue but it is likely to create confusion and potentially misleading.

The authors highlight that the changes in cognition observed within 15-min following exercise may not associated with exercise-induced increases in cortisol levels that may peak after the completion of cognitive assessment. This statement itself is fine but may be problematic in the context of introduction. For example, what are other faster exercise-induced physiological changes that could potentially be the mechanisms? In the reviewer’s opinion, it makes more sense to address what are the potential mechanisms that lead to the hypothesis on enhanced cognition immediately after exercise rather than point out what is not the potential mechanism (i.e., cortisol).

Round 2

Reviewer 3 Report

The review believe the revision should be accepted for publication.

This manuscript is a resubmission of an earlier submission. The following is a list of the peer review reports and author responses from that submission.

Round 1

Reviewer 1 Report

This is a well-written report on the immediate effects of exercise on behavioral and neurophysiological responses. The experimental design using repeated measures of the Flanker test (pre vs. post) and the rest vs. exercise protocols (counterbalanced order) for the same group of subjects is solid. The results contribute to the existing literature on the modulation effects of exercise on attentional conctrol and cognitive performance as well as the associated neurophysiological responses.

There are some technical issues regarding the statistical power and the analysis that could affect the proper interpretation of the data.

As the researchers acknowledged, the sample size of 11 subjects is very small. If 3 out of 11 subjects showed an opposite trend of pre-post changes (e.g. Figure 2), the statistical significance would probably be reduced to non-existent.  One simple remedy for strengthening the statistical power of the study is to increase the sample size to something like 20 or more subjects. If there is difficulty to do that due to limited resources, I would be good to split the test sessions into first half trials vs. second half trials to demonstrate consistency of the patterns.

Given the nature of multivariate behavioral and EEG measures from the same group of subjects, I think it is important to probe the relationship among the variables with mixed effects model with subject as a random effect to control for baseline differences among subjects (Koerner & Zhang, 2017). For instance, is there a significant relationship between the behavioral accuracy and behavioral reaction time as well as the N2, P3 responses? The predicted measure could be accuracy and the rest including ERP amplitude and latency data could be predictors. Age of the subjects could also be entered as a predictor as some studies show that P3 reflects an age effect. Koerner, T. K., & Zhang, Y. (2017). Application of linear mixed-effects models in human neuroscience research: A comparison with Pearson correlation in two auditory electrophysiology studies. Brain Sciences, 7, 26. There are a lot of online tutorials materials to learn how to implement mixed effects models.

The P3 amplitude and latency can be derived from the incongruent - congruent subtracted waveform rather than the ERPs for the individual type of stimuli. Would that change the results or not?

It is interesting to find out exactly what led to the absence or presence of significant effects in the behavioral and ERP data in the pre vs. post comparisons. In this regard, the sample size issue needs to be highlighted, and the brain-behavior correlation issue should be examined with more sophisticated models.

Reviewer 2 Report

The manuscript by Winneke and colleagues presents a study into the effects of an acute bout of moderate intensity exercise on attentional control using behavioral (accuracy and reaction time during a modified Eriksen flanker task) and neuroelectric measures (the amplitude and the latency of N2 and P3 components of event related potentials; ERPs) in healthy weight young adults. The study employed a within-subject cross-over design with posttest neurocognitive comparisons. The strengths of the study include neuroelectric assessment of brain function and individual calibration of the intensity of an exercise bout relative to 50-75% of participant’s maximal oxygen uptake. Unfortunately, the study suffers from major methodological limitations which prevent the reviewer from recommending it for publication in JCM. Specifically, the study does not include a standardized control condition. While neurocognitive assessments were conducted following 20 minutes of moderate intensity exercise, no true rest condition was included as during the non-exercise condition participants engaged in neurocognitive testing immediately following practice trials on the cognitive task (~33 seconds). This condition cannot be directly compared to the exercise condition as the activities that participants engaged in prior to cognitive testing were not standardized and remain unknown. The lack of the control group precludes meaningful conclusions from the study because day to day variability in cognitive performance was not adequately controlled for through the study design. For example, participants may have engaged in a number of different activities immediately before the non-exercise session, which would differentially influence their cognitive performance (e.g. studying, intellectually demanding work compared to TV viewing, active travel compared to driving to the session). Likewise, physical activity in the days preceding the testing sessions, food and caffeine intake, sleep and the time of day of each testing session were not controlled for, while these factors modulate baseline levels of cognitive performance. Unfortunately, these methodological concerns could only be addressed by repeating the study using a control group and by adequately controlling for the above-mentioned confounding factors. The reviewer would encourage authors to follow the recommendations outlined in the cited review by Pontifex and colleagues (2019) to guide future study design and the choice of a control group.

Another limitation of the study is a small sample size. While the study was powered to detect a large effect, meta-analytical findings indicate that the acute effects of moderate intensity exercise bouts on cognitive measures are small (e.g. d = 0.11 as reported by Chang et al. 2012). As such, the study would have 7% power to detect the small effect, assuming Cohen’s f = 0.055, n=11, α=0.05, 2 groups, 2 measures and correlation between repeated measures of 0.75 (G*Power v. 3.1.9.2, Faul, Erdfelder, Lang and Buchner, 2007). Thus, the null results in relation to cognitive measures are difficult to interpret. Authors are encouraged to present sample size calculations in future manuscripts.

Lastly, the significance and the novelty of the study were not clear given that the majority of studies focusing on the acute effects of exercise on cognitive functions measured cognitive performance immediately following the completion of an exercise bout (e.g. Pontifex et al. 2019 for the review of studies). The studies using neuroimaging were more evenly distributed in relation to the timing of neurocognitive assessment (approximately a third of studies included assessments immediately following the exercise bout). Likewise, the majority of research in this area has focused on young adults. From the public health perspective an important question remains on how long the effects of acute exercise on neurocognitive performance persist and what is the temporal trajectory of this effect. Few studies have attempted to answer this question using neuroimaging methods and they vary in populations of interest, neuroimaging methods and the delay of testing following exercise (e.g. Lambrick et al. 2016 in children using functional near infrared spectroscopy or McIntosh et. Al. 2014 in healthy adults using functional magnetic resonance imaging). Perhaps such assessment could be incorporated into the current study design in future reiterations of the study. The knowledge of temporal trajectory of neurofunctional responses following a single bout of moderate intensity exercise would help inform future interventions aimed at optimizing cognitive performance during a work day.

Reviewer 3 Report

This study examined the acute exercise effect on behavioral performance during a modified flanker task at baseline or immediately following exercise in adult participants. N2 and P3 components of ERP were assessed to examine the neuroelectrical activities associated with the information processing underlying the flanker task. The results showed no changes in behavioral outcomes but shorter P3 latency following exercise compared with baseline levels. The authors concluded that a single bout of exercise could benefit attentional control processes as manifested by faster information processing speed. Overall this is fine research that has practical implications. However, several major issues exist and require revision before this manuscript being considered for publication at this time.

Abstract

Line 15: What are the certain conditions?

Introduction

Although this is study provide evidence to replicate many of the prior research, it can hardly provide new information beyond what has been done in the field. The authors argue that limited studies examined immediate acute exercise effect on cognition and P3 but it is not clear why is it scientifically or practically important to investigate this specific time interval (i.e., within 5 min) as compared to others, which have been frequently studied in the literature.

The first paragraph the authors start with a statement that no consensus regarding acute exercise effect on cognition due to heterogeneity in study design. This study examined the immediate acute exercise effect but how can this study address the issue of discrepant research design? Also, this study used a flanker paradigm that is very different (i.e., decision-making process on congruency by color, both vertical and horizontal flanking effects) from the majority of previous research. This appears to create problems associated with heterogeneity in research design.

Line 63: P3 amplitude is typically used to reflect updating of mental representation in the working memory and attentional resources allocation. It is not clear why P3 amplitude reflects the intensity of processing. Do the authors suggest that larger P3 amplitude means more intense information processing? How to define the intensity of information processing?

Line 66: how about P3 amplitude during conflict trials?

Line 71: Why included Pontifex and Hillman study, which examined N2 during exercise? The mechanisms underlying the changes in cognitive performance and brain activity during and following exercise are likely different (see the RAH model in Dietrich et al. [2011]).

Line 96: Pontifex et al. (2015) actually found acute exercise effects on P3 amplitude but not P3 latency. Completely wrong description of the study.

Line 103: Do the authors believe that exercise-induced changes in cortisol levels are related to enhanced cognition following exercise? Any citation?

Method

Line 122: “that is comparable” appears twice in the sentence.

What was the stimulus-response mapping during the flanker task? For example, press a button for red color and the other for green color? What was the probability of green and red target?

Line 152-155: It is unclear how presenting flanking stimuli and central target simultaneously would prevent the prepotent response activation and inhibition. Unless the flanking stimuli were not processed at all (i.e., the subjects did not see them), they should signal the incorrect response activation, and thus require inhibition over such prepotent response.

Line 195: Based on ACSM’s guideline, an average of 63.58% HRmax should be considered as light intensity (<64% of HRmax). Did the authors use a different standard to define the intensity of exercise? Or this manuscript should be changed in to the acute effect of light-intensity exercise on cognition?

Did the authors record HR prior to the flanker task on the HR was not recorded during flanker task on the Rest-Condition day? Were the participants restricted from engaging in physical activity prior to testing or any substance that could influence their cognitive or physical function?

Result

In the method-statistic section, it is stated that 2 x 2 ANOVAs were to be used but an electrode site factor was included in all N2 and P3 analyses. Please be consistent.

Line 337: although it is possible that grand average waves do not reflect statistical results, the waves in figure 5 is likely to create confusion. Since the P3 latency effect was not interacted with Congruency, it may help to plot the ERP waves using data collapsed across congruent and incongruent trials. It is guaranteed that the plot would look better but it is worthy to try.

Discussion

Line 378: Pontifex et al. (2015) used a 3-stimulus oddball task, which is a simple P3 task. This task involves discrimination processes with little inhibition component as it does not present distraction that competes for attentional resources nor elicit prepotent incorrect response. Also, although that study did not show increased P3b amplitude from preexercise to postexercise, a decrease in P3b amplitude was found following rest. The study also showed larger P3b amplitude during postexercise test compared to postrest test. In brief, Pontifex et al. (2015) did not resemble the current study in many ways (i.e., experimental design, cognitive task, EEG data reduction, findings) The authors should make sure they have a complete understanding of their citations to avoid potential misleading statements.

If the main focus of this study is the timing of cognitive performance within a short delay following exercise, perhaps the authors could conduct analysis by task block. This approach may not result in enough EEG segments for each trial type in each task block but it may be worthy to try.

Line 403: since the limited statistical power is addressed here, why not provide a post-hoc analysis to determine the power of the current study to detect the acute exercise effect based on the effect sizes reported in the previous studies?

Line 428: Cognition during exercise is essentially a dual-task, which is different from measuring cognition following exercise. Also, findings regarding the modulation of cognitive performance and P3 have been inconsistent due to heterogeneity across studies. It becomes very confusing especially when the authors attempted to emphasize this notion in line 442. The readers may get lost in terms of what is the main point the authors are trying to convey in this paragraph.

A limitation section is required in the discussion section. In addition to sample size, many issues need to be addressed. For example, lack of a true control group, lack of pretest prior to experimental condition to detect the pre-post changes in cognitive performance and ERP indices, …etc.

It is good that the authors attempted to provide conclusions and implications of the current findings. However, this section could have been written in almost the same way if the acute exercise effect on cognition were to be examined after 10-min, 20-min, or longer. The authors should explicitly explain why it is important to investigate cognitive performance immediate after exercise throughout the manuscript and make conclusion accordingly.